# Research on Fault Tree Reconstruction Based on Contingency

**Song Xin** [1,2,3], **Xiaozhen Zhu** [1,2], **Shangxiao Liu** [1,2,*] and **Jianghui Guo** [1,2]

1   College of Safety and Environmental Engineering, Shandong University of Science and Technology, Qingdao 266590, China; xinsong@sdust.edu.cn (S.X.); zxz19971227@163.com (X.Z.); gjh199609@126.com (J.G.)
2   State Key Laboratory of Mining Disaster Prevention and Control Co-Found by Shandong Province and the Ministry of Science and Technology, Shandong University of Science and Technology, Qingdao 266590, China
3   College of Transportation, Shandong University of Science and Technology, Qingdao 266590, China
*   Correspondence: liushangxiao_24@163.com

**Abstract:** The fault tree analysis (FTA) method is an important analysis method for safety system engineering. Traditional accident analysis theory agrees that basic events lead to top events, but it does not fully consider that the accident process is accidental, and the calculation results exaggerate the probability of accident occurrence. This paper selects typical collision accidents, analyzes the shortcomings of the existing fault tree, indicates that there is a contingency in the accident process, constructs a probability fault tree based on the traditional fault tree, and puts forward concepts of "probability AND gate" and "probability OR gate". In addition, based on the traditional quantitative analysis method of fault trees, calculations of the occurrence probability, probability importance coefficient, and critical importance coefficient of top events are modified, and the modified quantitative calculation is applied to accident cases.

**Keywords:** fault tree reconstruction; contingency; probability AND gate; probability OR gate





## 1. Introduction

The causal analysis of accidents is a common method for identifying and analyzing the causes of accidents and preventing or controlling them. However, both the causal model and the causal analysis method have some shortcomings [1–4]. Using fault tree analysis, we can analyze the optimization decision, accident prediction, and accident investigation processing of the entire system, as well as discuss system safety [5,6]. Fault tree analysis is the quantitative analysis of a fault tree, that is, quantification of the top-event probability as the core goal and accurate expression of the accident risk degree with data [7].

According to the existing fault tree analysis method and the event-causal chain, basic cause events lead to intermediate events and then to top events. The theoretical description of a series of events leading to accidents according to a specific causal relationship is too simplistic [8,9], and different from reality.

Wang et al. [10] believe that if the nonquantifiable judgment of failure probability is insufficient, the logical relationship between all events cannot be measured. Their research results indicate that fault tree analysis cannot essentially deal with the dynamic process of accidents. Therefore, a new event tree analysis method including probability basic events was proposed. Hua et al. [11] considered the accidental explosion of dangerous goods in Tianjin Port, China, as the research object and systematically analyzed the causes of the accident based on fault tree analysis. Their results showed that the basic events of the fault tree should be introduced into the probability model, which should then be used to quantitatively analyze and judge various top events. Zhu et al. [12] reported that in fault tree analysis, with the extension of time, the impact of basic events on top events changes, that is, the probability of basic events changes with time. However, their research did not consider the contingency in the accident chain, which leads to a change in the impact of basic events on top events.

Many studies have recommended different methods of dealing with uncertainty in FTA, including, but not limited to, fuzzy set theory [13] and the Bayesian network [14]. Mohammad Yazdi [15] reviewed the uncertainty treatment in risk assessment based on fault tree analysis (FTA) in the past decade and found that the research on uncertainty treatment in qualitative and quantitative risk assessment is a developing field. Because the logical relationship of the fault tree is artificially determined, the contingency in the process of transferring basic events to top events in the fault tree is ignored.

In response to the above problem, this paper considers the chance nature of accidents to analyze and explain the existing problems of a typical fault tree. Based on this, the concepts of the probability AND gate and the probability OR gate are proposed, the fault tree is modified, the concept of a probability fault tree is proposed, and the quantitative calculation method is improved. This theory is helpful to produce a more reasonable risk assessment of fault tree analysis.

## 2. Analysis of the Construction Process of a Probability Accident Tree

### 2.1. Problems in the Existing Fault Tree

To analyze problems in the existing fault tree, this section selects typical traffic accidents for collision accident analysis and combined with the accident contingency, analyzes accident occurrence and development, identifies the possible problems, and lays the foundation for the transformation of the fault tree (Figure 1).

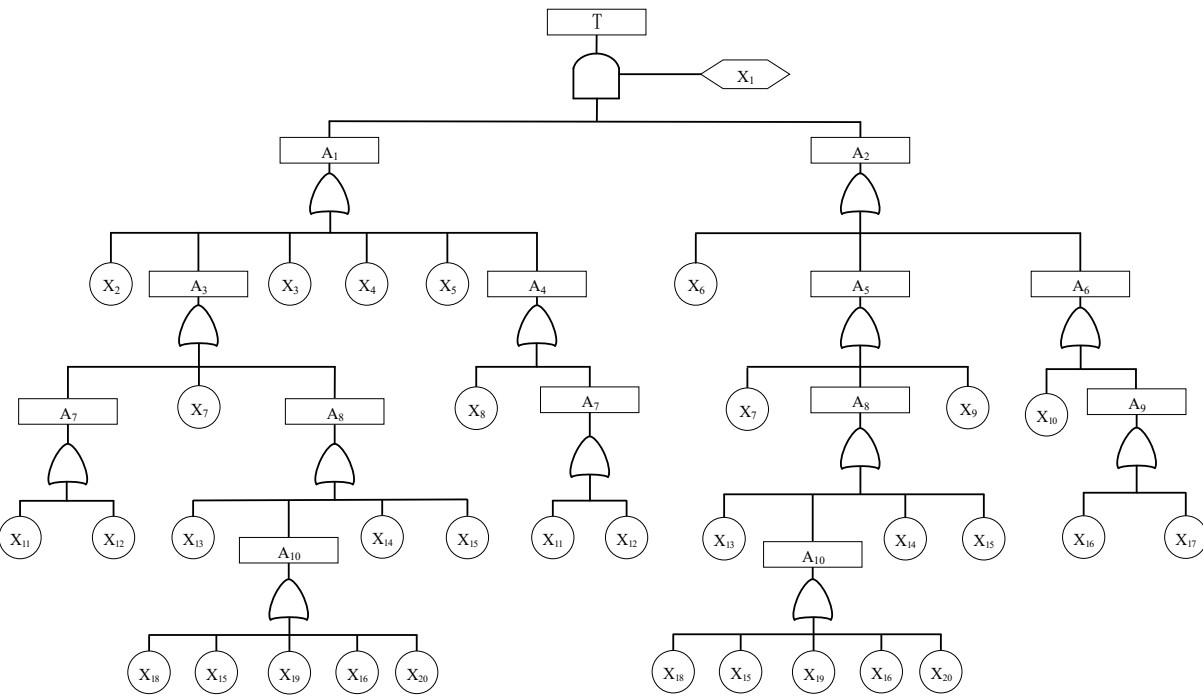

**Figure 1.** Fault tree of a collision accident.

In order to facilitate the subsequent analysis of the evolutionary fault tree, each event is generally marked with a word symbol. The top event is represented by T, and the intermediate event is represented by A, which is distinguished from the basic event X in the fault tree, as shown in Table 1.

According to the existing collision fault tree and event chain, during accident occurrence and development, the basic events or a combination of basic events leads to the occurrence of the accident layer by layer; that is, previous conditions inevitably lead to the occurrence of subsequent events, such as poor road traffic conditions leading to fatigue driving, which, in turn, leads to inattention, incorrect judgment, insufficient longitudinal distance, and too fast a speed, which further leads to collisions.

**Table 1.** Fault tree accident type and symbol comparison [16].

| Symbol | Event Name | Symbol | Event Name | Symbol | Event Name |
|---|---|---|---|---|---|
| T | Collision accident | X1 | Inability to dodge | X12 | Poor lighting for driving at night |
| A1 | Insufficient vertical spacing | X2 | Illegal turn | X13 | Bad mood |
| A2 | Speed too fast | X3 | Violation of giving way to an oncoming vehicle | X14 | Drive after drinking |
| A3 | Misjudgment | X4 | Overtaking in violation of regulations | X15 | Poor health |
| A4 | Observation error | X5 | Illegal parking | X16 | Poor road traffic conditions |
| A5 | Operation error | X6 | Pursuit of stimulation | X17 | Bad brake performance or too much wear |
| A6 | Braking problem | X7 | Lack of knowledge and experience | X18 | Lack of sleep |
| A7 | Poor sight | X8 | Not enough sight distance | X19 | Driving too long |
| A8 | Inattention | X9 | Limited driving skills | X20 | Poor temperature ventilation |
| A9 | Poor braking | X10 | Brake failure | | |
| A10 | Fatigue driving | X11 | Impact of rain, snow, and fog | | |

Previous studies have found that there is a certain contingency in this chain of events, and the occurrence of accidents is not inevitable. For example, poor road traffic conditions do not necessarily cause fatigue driving, and a poor mood does not necessarily lead to a lack of concentration. For knowledge-level experience, insufficiency does not necessarily cause judgment errors. Therefore, the accident tree has certain loopholes; it ignores the contingency of the event itself and the event chain in the transmission process, that is, the probability of the AND gate or the OR gate. The description of the causality in accident development is too absolute, and it is believed that an accident is the inevitable result of the causal transmission of various factors. Therefore, the existing accident tree should be improved so that accidents can be fully understood.

### 2.2. Fault Tree Reconstruction

Analysis of typical collision accidents shows that there is a contingency in accident occurrence and development, that is, in event tree construction, which was not considered by the traditional AND and OR gates. To fully understand accidents, the concepts of the probability AND gate, probability OR gate, and probability fault tree are proposed.

#### 2.2.1. Probability AND Gate

The probability AND gate mean that when input events, $B_1$ and $B_2$, occur simultaneously, output event $A$ may not necessarily occur. Based on the input events, $B_1$ and $B_2$, occurring simultaneously, there may be chance events that prevent output event $A$ from occurring. Therefore, there is a probability that when input events, $B_1$ and $B_2$, occur simultaneously, output event $A$ occurs. Namely, $A = B_1 \cap B_2 \cap B_{12}$, or $A = B_1 \cdot B_2 \cdot B_{12}$. This is also true if multiple input events exist, such as $A = B_1 \cdot B_2 \cdots B_n \cdot B_{12 \ldots n}$. The probability AND gate symbol is shown in Figure 2.

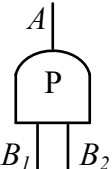

**Figure 2.** Probability AND gate symbol.

#### 2.2.2. Probability OR Gate

The probability OR gate means that at least one of the input events, $B_1$ and $B_2$, occurs and output event $A$ does not necessarily occur. Based on at least one of the input events, $B_1$ and $B_2$, there may be chance events that prevent output event $A$ from occurring. Therefore, there is a probability that when at least one of the input events, $B_1$ and $B_2$, occurs, output event $A$ will occur. Namely, $A = B_1 B_{m_1} \cup B_2 B_{m_2}$, or $A = B_1 B_{m_1} + B_2 B_{m_2}$. This is also true for multiple input events. The probability OR gate symbol is shown in Figure 3.

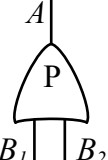

**Figure 3.** Probability OR gate symbol.

2.2.3. Probability Accident Tree

Through an analysis of the probability AND and probability OR gates, the probability accident tree was constructed, as shown in Figure 4. Probability fault trees reflect that there is a specific chance of occurrence of the top event. Therefore, the probability AND and probability OR gates are used to connect basic events, intermediate events, and top events.

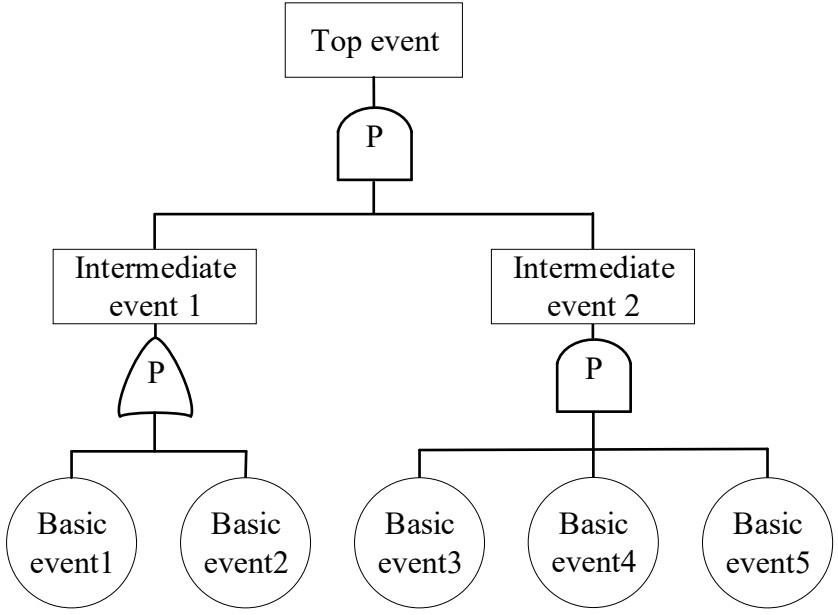

**Figure 4.** Schematic diagram of probability accident tree.

When at least one of basic events 1 and 2 occurs, it does not necessarily lead to the occurrence of intermediate event 1. There is a probability $q_{m_r}$ that when at least one of basic events 1 and 2 occurs, intermediate event 1 occurs. That is, there is a probability $\overline{q_{m_r}}$ that when at least one of the basic events 1 and 2 occurs, intermediate event 1 does not occur. If basic event 1 inevitably leads to intermediate event 1, $q_{m_r} = 1$ and $\overline{q_{m_r}} = 0$.

**3. Improvement of the Quantitative Calculation Method of Fault Trees**

Quantitative analysis of the accident tree is mainly based on the occurrence probability of each basic event, calculation of the top event's occurrence probability, and probability and the critical importance of each basic event [17]. The concept of the probability AND and probability OR gates is introduced after the above modification of the accident tree; therefore, the corresponding quantitative analysis method of the accident tree also needs to be improved.

*3.1. Improvement of the Quantitative Calculation Method*

The calculation of the top event's occurrence probability is the basis of the fault tree's quantitative analysis [18], and the improved method for calculating the top event's occurrence probability considers the contingency in the transmission process of the event chain; thus, the resulting probability of the accident is closer to reality.

The probability product of events connected by the probability AND gate is

$$q_A = q_j \prod_{i=1}^{n} q_i \tag{1}$$

where

$$\prod_{i=1}^{n} q_i = q_1 q_2 \cdots q_n \tag{2}$$

and

$$q_j = 1 - \overline{q_j} \tag{3}$$

In this formula, $q_j$ is the occurrence probability of the $i$-th basic event, $q_A$ is the probability of AND gate events, $n$ is the number of input events, $q_j$ is the probability that the occurrence of the basic event leads to the occurrence of an AND gate event, $\overline{q_j}$ is the probability that the occurrence of the basic event does not lead to the occurrence of an AND gate event, and $\prod$ is a mathematical operation symbol that indicates the product of the probabilities.

The sum of the probabilities of events connected by a probability OR gate is

$$q_o = \coprod_{i=1,m=1}^{n} q_i q_m = 1 - \prod_{i=1,m=1}^{n} (1 - q_i q_m) \tag{4}$$

where

$$q_m = 1 - \overline{q_m} \tag{5}$$

In this formula, $q_o$ is the probability of OR gate events, $q_m$ is the probability that the occurrence of the basic event leads to the occurrence of an OR gate event, $\overline{q_m}$ is the probability that the occurrence of the basic event does not lead to the occurrence of an OR gate event, and $\coprod$ is a mathematical operation symbol that indicates the sum of the probabilities.

The minimum cut set is used to calculate the occurrence probability of the top events, and there are no repeated events.

$$g = \coprod_{r=1}^{k} (q_{j_r} \prod_{x_i \in k_r, j=1}^{k} q_i) q_{m_r} = 1 - \prod_{i=1}^{n} \left( 1 - q_m q_j \prod_{x_i \in k_r} q_i \right) \tag{6}$$

In this formula, $x_i$ is the $i$-th basic event, $k_r$ is the $r$-th minimum cut set, $k$ is the number of minimum cut sets, $x_i \in k_r$ is the $i$-th basic event belonging to the $r$-th minimum cut set, $q_{j_r}$ is the probability of the occurrence of $k_r$ caused by the occurrence of the basic event, and, $q_{m_r}$ is the probability that the occurrence of $k_r$ leads to an accident.

If there are repeated events in the minimum cut set, the occurrence probability of the top event is

$$g = \sum_{r=1}^{k} q_{j_r} q_{m_r} \prod_{x_i \in k_r} q_i - \sum_{1 \le r \le s \le k} q_{j_r} q_{j_s} q_{m_r} q_{m_s} \prod_{x_i \in k_r \cup k_s} q_i + \cdots$$
$$+ (-1)^{k-1} \prod_{r=1, x_i \in k_r}^{k} q_i q_{j_r} q_{m_r} \tag{7}$$

In this formula, $r, s$ is the sequence number of the smallest cut set.

The occurrence probability of the event at the top of the accident tree has been modified above; therefore, the probability importance and critical importance of the corresponding basic events should also be modified and studied. Among these, Equations (8) and (9), the probabilistic importance coefficient and the critical importance coefficient, remain unchanged. However, if the top event's probability changes, then the magnitudes of the probabilistic and critical importance coefficients also change accordingly.

$$I_g(i) = \frac{\partial g}{\partial q_i} \tag{8}$$

In this formula, $I_g(i)$ is the probability importance coefficient of a basic event $x_i$.

$$CI_g(i) = \frac{q_i}{g} I_g(i) \tag{9}$$

In this formula, $CI_g(i)$ is the critical importance coefficient of a basic event $x_i$.

### 3.2. Application of the Probability Fault Tree

With the development of the urban economy, the number of high-rise buildings in cities is gradually increasing. At the same time, fires in these high-rise buildings have become the focus of firefighting. This section describes the construction of a probability accident tree and draws a probability accident tree by analyzing fire accidents in the college and university dormitories [19]. The occurrence probability of each basic event in the probability accident tree is recorded in Table 2 [20].

**Table 2.** Probability accident tree accident event type and occurrence probability of basic events.

| Symbol | Event Name | Probability Value | Symbol | Event Name | Probability Value |
|---|---|---|---|---|---|
| T | Dormitory fire accident | | A6 | Open fire source | |
| A1 | On fire | | A7 | Electric fire source | |
| A2 | Out of control | | A8 | Firefighting facilities not working | |
| A3 | Fire source | | A9 | Line | |
| A4 | Combustible | | A10 | Electrical appliances | |
| A5 | Rescue not timely | | | | |
| X1 | Smoking | 0.02 | X13 | Clothes, quilt | 0.02 |
| X2 | Alcohol stove | 0.05 | X14 | Textbook, desk | 0.03 |
| X3 | Burning mosquito repellent incense | 0.02 | X15 | Other flammable materials | 0.02 |
| X4 | Thunder | 0.03 | X16 | Alarm system not working | 0.05 |
| X5 | Connecting wires without permission | 0.05 | X17 | Not found in time | 0.03 |
| X6 | Wires short-circuited | 0.02 | X18 | At a loss | 0.03 |
| X7 | Line aging | 0.03 | X19 | Fire extinguishers in short supply | 0.05 |
| X8 | Use of illegal electrical appliances | 0.06 | X20 | Insufficient input in fire hydrants | 0.01 |
| X9 | Use of inferior electrical appliances | 0.05 | X21 | Unreasonable configuration of the automatic sprinkler system | 0.02 |
| X10 | Use of an electric heater | 0.03 | X22 | Lack of knowledge of how to use a fire extinguisher | 0.03 |
| X11 | Use of an electric blanket | 0.03 | X23 | Insufficient water supply for fire hydrants | 0.03 |
| X12 | Use of hand warmers | 0.02 | X24 | Failure of the automatic sprinkler system | 0.02 |

By simplifying the fault tree shown in Figure 5 and using the calculation for the probability of the event at the top of the fault tree in the early stage, we calculated the probability of fire in the dormitory as 0.98%.

Considering the uncertainty that exists in the accident process, the occurrence of a specific basic event does not necessarily lead to an intermediate event. The existence of the probability of occasionality allows the occurrence of a basic event without the occurrence of an intermediate event regardless of the size of that probability; using the revised fault tree quantitative calculation method, the calculated probability of a college dormitory fire is always ≤0.98%.

These results show that the accident probability obtained through the quantitative analysis of the original accident tree does not consider the contingency. We believe that a specific basic event would inevitably lead to intermediate events, thus leading to accidents. Therefore, the accident probability obtained was extremely high. In reality, contingencies exist in the occurrence of accidents, especially for a single accident, in which the contingency is more obvious. Therefore, the modified quantitative calculation method for accident trees has practical theoretical value.

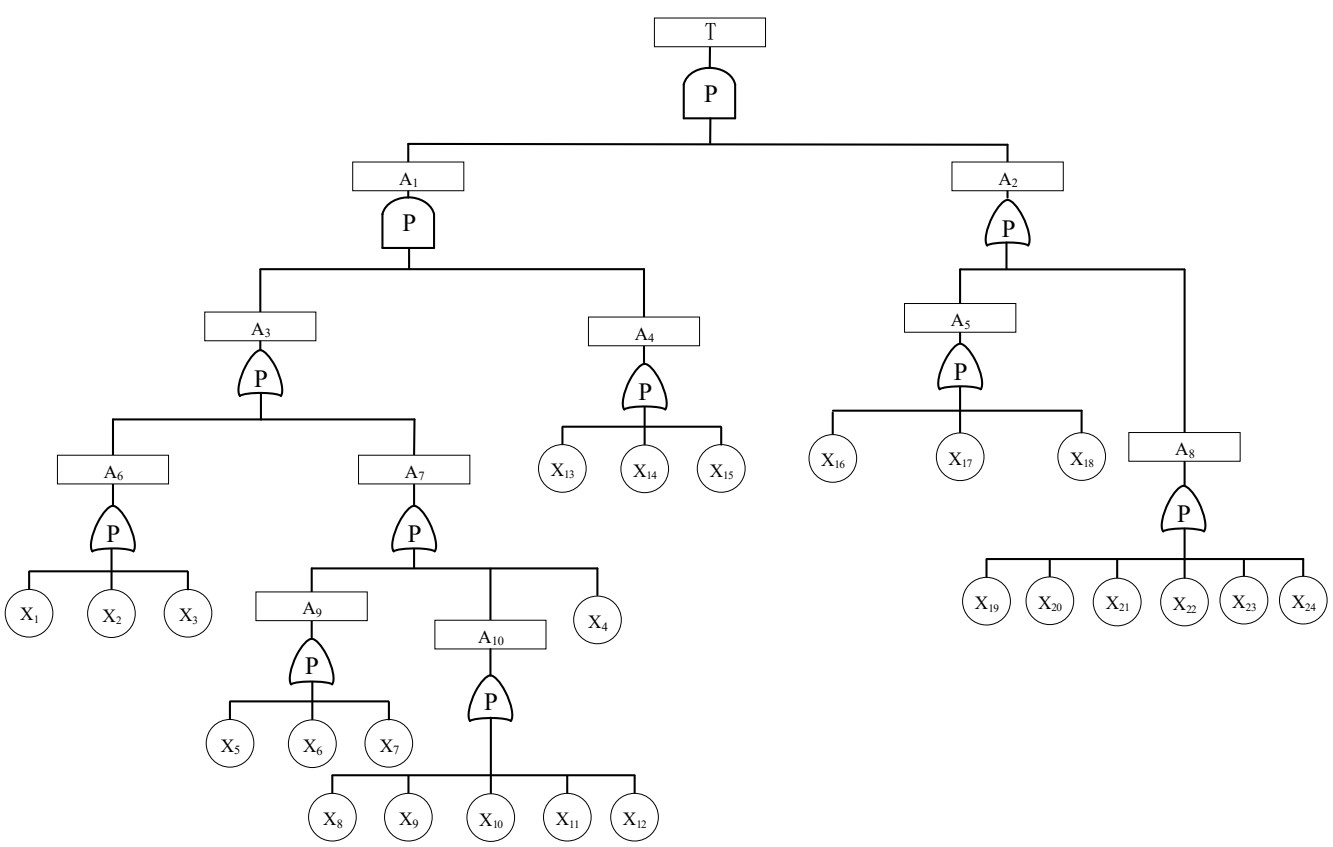

**Figure 5.** Fire accident tree diagram of a college dormitory.

## 4. Conclusions

(1) Through the analysis of typical collision accidents, the shortcomings of the existing FTA are analyzed, and the concept of a probabilistic FTA is innovatively proposed to enrich accident tree analysis. The concepts of probability AND and OR gates are presented and applied to the accident tree.

(2) The quantitative calculation method of traditional FTA essentially exaggerates the probability of an accident. Therefore, the traditional FTA has been reformed and the probability accident tree is compiled, and the quantitative calculation method of probability FTA is proposed.

(3) The proposal of probabilistic FTA has important practical significance and theoretical value for guiding safety management. It provides a new idea for the study of top event probability in traditional FTA.

**Author Contributions:** S.X. and S.L. presided over the main work and wrote the thesis; X.Z. completed the basic theoretical research; J.G. completed the data processing and analysis; they all provided insightful suggestions and revised the thesis. All authors have read and agreed to the published version of the manuscript.

**Funding:** This work has been funded by the National Natural Science Foundation of China, grant numbers 51774197.

**Institutional Review Board Statement:** This study was abandoned for ethical review and approval.

**Informed Consent Statement:** Informed consent was obtained from all subjects involved in the study.

**Data Availability Statement:** The study did not report any data.

**Conflicts of Interest:** The authors declare no conflict of interest.

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
