# Peer review of "Research on Fault Tree Reconstruction Based on Contingency"

_processes, doi:10.3390/pr10020427_

Round 1

Reviewer 1 Report

refer to review in pdf document

Author Response

尊敬的审稿人:

Thank you for arranging a timely review for our manuscript (ID: processes-1563490). According to the suggestion of the reviewer, we chose MDPI language editing service (ID:english-39765) to modify the English language and style of the article. In addition, We have carefully evaluated the reviewers' critical comments and thoughtful suggestions, responded to these suggestions point-by-point, and revised the manuscript accordingly. All changes made to the text are in red so that they may be easily identified. We appreciate for Editors/Reviewers' warm work earnestly and hope that the correction will meet with approval.

With regard to the editor and reviewers' comments and suggestions, we wish to reply as follows:

Review1:

The manuscript is clear, well written. The example is adequate and provides a good illustration of the proposed methodology. The manuscript can be accepted after minor revision without further reviewing.

Comment 1:

Line 157-158: the reference to equations (2.8) and (2.9) is incorrect in the context of this manuscript.

Answer 1:

Among these, equations (8) and (9), the probabilistic importance coefficient and critical importance coefficient, remain unchanged.

Comment 2:

“allows the for theoccurrence”中缺少一个单词

Answer 2:

The existence of the probability of occasionality allows the occurrence of the basic event without the occurrence of an intermediate event, regardless of size of that probability; using the revised fault tree quantitative calculation method, the calculated probability of a college dormitory fire is always less than or equal to 0.98%.

Comment 3:

第 191 行:没有引用。

答案3:

随着城市经济的发展,城市中高层建筑的数量逐渐增多。与此同时,高层建筑火灾也成为消防的重点。本节描述概率事故树的构建,并通过分析高校宿舍火灾事故[18]绘制概率事故树。概率事故树中各基本事件的发生概率见表2[19] ]。

Reviewer 2 Report

Reviewer’s comments on processes-1563490

“Research on fault tree reconstruction based on contingency”

The manuscript studies the fault tree analysis (FTA) considering the contingencies of accident occurrences in safety system engineering. The reviewer would like to suggest the following comments for the authors to consider and revise their manuscripts based upon:

  • Abstract should be improved. It should summarize the main achievements/findings of the research.
  • Introduction should be expanded. The research gap is not well defined. Many existing papers and industry practices about FTA are missing from the literature review.
  • The literature review only gives some papers and doesn’t provide details about what the conclusions of the papers were. The authors should better outline the key novelty of the paper compared to the extensive literature available.
  • Section 2 should be dedicated to theories of the idea. So it’s suggested to shorten the basic materials (which an be found in many other textbooks and papers) and focus on the proposed idea.
  • The section 3 reads very incomplete. The authors should test their idea on a numerical (real-life) case study and make very rich discussions about their results. Making comparisons between the results (with other studies) can be useful for readers to better understand the efficacy of the proposed method.
  • Reference list can be expanded. The authors can add some more recent publications including some papers from MDPI Processes.

Author Response

Dear Reviewer:

Thank you for arranging a timely review for our manuscript (ID: processes-1563490). According to the suggestion of the reviewer, we chose MDPI language editing service (ID:english-39765) to modify the English language and style of the article. In addition, We have carefully evaluated the reviewers' critical comments and thoughtful suggestions, responded to these suggestions point-by-point, and revised the manuscript accordingly. All changes made to the text are in red so that they may be easily identified. We appreciate for Editors/Reviewers' warm work earnestly and hope that the correction will meet with approval.

With regard to the editor and reviewers' comments and suggestions, we wish to reply as follows:

Review 2:

The manuscript studies the fault tree analysis (FTA) considering the contingencies of accident occurrences in safety system engineering. The reviewer would like to suggest the following comments for the authors to consider and revise their manuscripts based upon:

Comment 1:

Abstract should be improved. It should summarize the main achievements/findings of the research.

Answer 1:

Thank you for your suggestion. As you said, the abstract should reflect the main results of this study, which will help readers understand. The improved abstract is as follows:

Fault tree analysis method is one of the important analysis methods in safety system engineering. The traditional accident analysis theory agrees that the occurrence of basic events will lead to the occurrence of top events, but it does not fully consider that the accident process is accidental, and the calculation results exaggerate the probability of accident occurrence. This paper selects typical collision accidents, analyzes the shortcomings of the existing fault tree, points out that there is contingency in the process of accident, constructs the probability fault tree based on the traditional fault tree, and puts forward the concepts of "probability probability AND gate" and "probability OR gate". In addition, based on the traditional quantitative analysis method of fault tree, the calculation of the occurrence probability, probability importance coefficient and critical importance coefficient of top events are modified, and the modified quantitative calculation method is applied to accident cases.

Comment 2:

Introduction should be expanded. The research gap is not well defined. Many existing papers and industry practices about FTA are missing from the literature review.

Answer 2:

Thank you for your advice. The revised manuscript expands the introduction, adds references related to FTA, and clarifies the differences between the research content of this paper and that of other scholars. The revised manuscript is as follows:

This paper points out that there is contingency in the process of accidents and there is a certain contingency in the process of event chain, which is consistent with that reported by Wang et al. [11] Hua et al. [12] and Zhu et al.[13]

Wang et al. believes that if the non quantifiable judgment of failure probability is not sufficient, the logical relationship between all events can not be measured. Their research results believe that fault tree analysis can not essentially deal with the dynamic process of accidents. Therefore, a new event tree analysis method including probability basic events is proposed.

Hua et al. Took the explosion accident of dangerous goods in Tianjin port, China as the research object, and systematically analyzed the causes of the accident based on the fault tree analysis method. Their research results show that the basic events of the fault tree should be introduced into the probability model, and the probability model should be used to quantitatively analyze and judge various top events.

Zhu et al. Believe that in the fault tree analysis method, with the extension of time, the impact of basic events on top events will change, that is, the probability of basic events will change with time. However, their research does not consider the contingency in the accident chain, which leads to the change of the impact of basic events on top events.

Relevant references are attached:

[11]Wang W , Jiang X , Xia S , et al. Incident tree model and incident tree analysis method for quantified risk assessment: An in-depth accident study in traffic operation[J]. Safety Science, 2010, 48(10):1248-1262.

[12]HuaW, ChenJ S, QinQ, et al. Causation analysis and governance strategy for hazardous cargo accidents at ports: Case study of Tianjin Port's hazardous cargo explosion accident[J]. Marine Pollution Bulletin, 2021, 30(3):542-555.

[13]Zhu C , Tang S , Li Z , et al. Dynamic study of critical factors of explosion accident in laboratory based on FTA[J]. Safety Science, 2020, 130:104877.

Comment 3:

The literature review only gives some papers and doesn’t provide details about what the conclusions of the papers were. The authors should better outline the key novelty of the paper compared to the extensive literature available.

Answer 3:

Many literatures put forward different methods to deal with uncertainty in FTA, including but not limited to fuzzy set theory [14] and Bayesian network [15]. Hu et al.[16] Reviewed the uncertainty treatment in risk assessment based on fault tree analysis (FTA) in the past decade, and believed that the research on uncertainty treatment in the process of qualitative and quantitative risk assessment will be a developing field. Because the logical relationship of the fault tree is determined artificially, the contingency in the process of transferring the basic events to the top events in the fault tree is ignored.

In response to above problem, this article considers the chance nature of accidents to analyze and explain the existing problems of a typical fault tree. On this basis, the concepts of the probability AND gate and probability OR gate are proposed, the fault tree is modified, the concept of a probability fault tree is proposed, and the quantitative calculation method is improved. This theory is helpful to make a more reasonable risk assessment of fault tree analysis method.

Relevant references are attached:

[14]Kang J ,  Sun L ,  Soares C G . Fault Tree Analysis of floating offshore wind turbines[J]. Renewable Energy, 2019, 133(APR.):1455-1467.

[15]Chu Z , Yang Z , Peng M , et al. Research of security analysis based on subjective Bayesian Networks. IEEE, 2011.

[16]My A , Skb C , Mw B . Uncertainty handling in fault tree based risk assessment: State of the art and future perspectives[J]. Process Safety and Environmental Protection, 2019, 131:89-104.

Comment 4:

Section 2 should be dedicated to theories of the idea. So it’s suggested to shorten the basic materials (which an be found in many other textbooks and papers) and focus on the proposed idea.

Answer 4:

Thank you for your advice. We note that the explanation in section 2.2.3 is repeated, and the revised manuscript shortens the explanatory content. In addition, the structure of the fault tree starts from the basic event and expands gradually to the top event. There is contingency in each accident chain. Section 2 mainly introduces the construction process of probabilistic fault tree, including the process of "probability AND gate", "probability OR gate" and the influence of contingency on the occurrence of top events.

Comment 5:

The section 3 reads very incomplete. The authors should test their idea on a numerical (real-life) case study and make very rich discussions about their results. Making comparisons between the results (with other studies) can be useful for readers to better understand the efficacy of the proposed method.

Answer 5:

Thank you for your advice. Akin to the calculation principle used in the tradition FTA, the calculation, and simplifification, method of Boolean algebra, other methods and structural importance analysis also can qualitatively analyse an probabilistic fault tree is proposed in this paper. As the fire accidents of university dormitories listed in Section 3, according to the traditional fault tree calculation method, the probability of fire in university dormitories is 0.98%. However, as we have questioned, there is contingency in the process of the occurrence of basic events affecting top events, that is, the occurrence of a basic event does not necessarily lead to intermediate events. The probability of contingency makes the intermediate events not occur. Even if the probability of existence is very small, it will also affect the probability of occurrence of top events. Therefore, using the modified fault tree quantitative calculation method, the calculated probability of college dormitory fire is always less than or equal to 0.98%.

At present, most of the probability values of basic events are based on investigation and statistics, and there is no definite probability value, and at present, the contingency probability that may lead to the non occurrence of intermediate events (top events) can only be estimated according to the frequency of many basic events. Therefore, it is impossible to accurately calculate the accurate value of the probability of basic events considering contingency. It is worth noting that considering the contingency, the probability value of intermediate event (top event) is always less than or equal to the probability value calculated by the current traditional fault tree. In other words, the traditional fault tree calculation results exaggerate the probability of accidents, which is the question raised in this paper.

Comment 6:

Reference list can be expanded. The authors can add some more recent publications including some papers from MDPI Processes.

Answer 6:

Thank you for your suggestion. The revised manuscript increases the number of references, including relevant papers published in recent years.

Reviewer 3 Report

The authors seem totally unaware of the literature on fault tree analysis and more generally on reliability engineering.

Their "proposal" consists eventually in adding a basic event to AND and OR gates so to take into account some uncertainty in the phenomena. 

The calculation methods they are using are outdated.

Author Response

Thank you for arranging a timely review for our manuscript (ID: processes-1563490). According to the suggestion of the reviewer, we chose MDPI language editing service (ID:english-39765) to modify the English language and style of the article. In addition, We have carefully evaluated the reviewers' critical comments and thoughtful suggestions, responded to these suggestions point-by-point, and revised the manuscript accordingly. All changes made to the text are in red so that they may be easily identified. We appreciate for Editors/Reviewers' warm work earnestly and hope that the correction will meet with approval.

With regard to the editor and reviewers' comments and suggestions, we wish to reply as follows:

Review 3

The authors seem totally unaware of the literature on fault tree analysis and more generally on reliability engineering.

Comment :

Their "proposal" consists eventually in adding a basic event to AND and OR gates so to take into account some uncertainty in the phenomena. The calculation methods they are using are outdated.

Answer:

Thank you for your question.

The theory and method of fault tree analysis are not invariable. Because the logical relationship of fault tree is determined artificially, it ignores the contingency in the process of transferring the basic events to the top events in the fault tree. Based on this, this paper improves the traditional fault tree analysis method and considers the influence of contingency in the fault tree analysis method. This method considers the possibility of top events caused by basic events and the uncertainty of impact. It can also support a more reasonable risk assessment of the fault tree analysis method. As the fire accidents of university dormitories listed in Section 3, according to the traditional fault tree calculation method, the probability of fire in university dormitories is 0.98%. However, as we have questioned, there is contingency in the process of the occurrence of basic events affecting top events, that is, the occurrence of a basic event does not necessarily lead to intermediate events. The probability of contingency makes the intermediate events not occur. Even if the probability of existence is very small, it will also affect the probability of occurrence of top events. Therefore, using the modified fault tree quantitative calculation method, the calculated probability of college dormitory fire is always less than or equal to 0.98%.

In our previous research[1], we proposed "Rzeconstruction of the fault tree based on accident evolution". Considering that the top event may continue to evolve into further accidents, it is verified that the traditional quantitative calculation method of fault tree is still applicable. In this study, we consists that in adding a basic event to AND and OR gates so to take into account some uncertainty in the phenomena,akin to the calculation principle used in the tradition FTA, the calculation, and simplifification, method of Boolean algebra, other methods and structural importance analysis also can qualitatively analyse an probabilistic fault tree.

[1]Xin S, Zhang L, Jin X, et al. Reconstruction of the fault tree based on accident evolution - ScienceDirect[J]. Process Safety and Environmental Protection, 2019, 121:307-311.

Round 2

Reviewer 2 Report

The authors have implemented the comments. The only last suggestion I would like to make to authors is to make another proofread and make sure the manuscript is free of typos.

Author Response

Thank you for your advice. We checked the full text to make sure there were no typos.

Reviewer 3 Report

The answers provided by the authors are not convincing and show their lack of knowledge about basic reliability theory.

Author Response

Thank you for your advice.

FTA is an effective and forward-looking tool for analyzing system safety and reliability. Based on standard FTA, several extensions are proposed, such as dynamic fault tree, event fault tree, time fault tree, etc. [1].Many studies [2-3] show that the traditional fault tree calculation result is not an accurate value. The limitation of fault tree is to assume that the fault probability of basic events is accurately known, but in real accidents, the basic events are relatively complex, so it is difficult to give an occurrence probability.Traditional FTA requires the logical relationship between events and event probability values, which can only represent the state of "occurrence / non occurrence", which makes FTA almost impossible to analyze the reliability of uncertain systems [4-5]. However, uncertainty is an important factor in risk assessment [6].

Other scholars have made efforts to transform the traditional FTA. Bi et al. [7] proposed a new method combining T-S fuzzy fault tree and Bayesian network based on the ability of Bayesian network to process uncertain information. The results are more accurate than traditional analysis methods. Yin et al. [8] used the method of combining SAM and fuzzy set theory to deal with different forms of opinions of different experts, reasonably convert the opinions of multiple experts into a certain probability value, and determine the more reasonable probability of the basic event of the fault tree, but it only improved the probability of the basic event, without considering the possibility and influence uncertainty of the top event caused by the basic event.

The theory and method of FTA are not invariable. Because the logical relationship of fault tree is determined artificially, it ignores the contingency in the process of transferring the basic events to the top events in the fault tree.According to the existing fault tree analysis method and event causal chain, the basic cause event leads to the intermediate event, and then the top-level event. However, the theoretical description of a series of events leading to accidents according to specific causality is too simple [9-11], which is different from reality. The idea proposed in this paper is to consider that there is contingency in the process of accident occurrence and certain contingency in the process of event chain, which leads to uncertainty in traditional FTA analysis.

Based on this, this paper improves the traditional fault tree analysis method and considers the influence of contingency in the fault tree analysis method. This method considers the possibility of top events caused by basic events and the uncertainty of impact. It can also support a more reasonable risk assessment of the fault tree analysis method.In our previous research[12], we proposed "Rzeconstruction of the fault tree based on accident evolution". Considering that the top event may continue to evolve into further accidents, it is verified that the traditional quantitative calculation method of fault tree is still applicable. In this study, we consists that in adding a basic event to AND and OR gates so to take into account some uncertainty in the phenomena,akin to the calculation principle used in the tradition FTA, the calculation, and simplifification, method of Boolean algebra, other methods and structural importance analysis also can qualitatively analyse an probabilistic fault tree.

Kabir S . An overview of fault tree analysis and its application in model based dependability analysis[J]. Expert Systems with Applications, 2017, 77:114-135.

Kumar M , Kaushik M . System failure probability evaluation using fault tree analysis and expert opinions in intuitionistic fuzzy environment[J]. Journal of Loss Prevention in the Process Industries, 2020, 67(2):104236.

Purba J H , Tjahyani D , Ekariansyah A S , et al. Fuzzy probability based fault tree analysis to propagate and quantify epistemic uncertainty[J]. Annals of Nuclear Energy, 2015, 85(NOV.):1189-1199.

Bba B , Cx A , Xl A , et al. Application of integrated factor evaluation–analytic hierarchy process–T-S fuzzy fault tree analysis in reliability allocation of industrial robot systems. 2021.

Khakzad N , Khan F , Amyotte P . Safety analysis in process facilities: Comparison of fault tree and Bayesian network approaches[J]. Reliability Engineering & System Safety, 2011, 96(8):925-932.

Hu L , Kang R , Pan X , et al. Risk assessment of uncertain random system—Level-1 and level-2 joint propagation of uncertainty and probability in fault tree analysis[J]. Reliability Engineering [?] System Safety, 2020, 198:106874.

Bi Z , Li C , Li X , et al. Research on Fault Diagnosis for Pumping Station Based on T-S Fuzzy Fault Tree and Bayesian Network[J]. Journal of Electrical and Computer Engineering, 2017, 2017:1-7.

Hy A , Cl B , Wei W B , et al. Safety assessment of natural gas storage tank using similarity aggregation method based fuzzy fault tree analysis (SAM-FFTA) approach[J]. Journal of Loss Prevention in the Process Industries, 2020, 66.

Khan F I , Iqbal A , Ramesh N , et al. SCAP: a new methodology for safety management based on feedback from credible accident-probabilistic fault tree analysis system[J]. Journal of Hazardous Materials, 2001, 87(1-3):23-56.

Faisal I Khan,Asad Iqbal,N Ramesh,S.A Abbasi. SCAP: a new methodology for safety management based on feedback from credible accident-probabilistic fault tree analysis system[J]. Journal of Hazardous Materials,2001,87(1):23-56.

Wang W , Jiang X , Xia S , et al. Incident tree model and incident tree analysis method for quantified risk assessment: An in-depth accident study in traffic operation[J]. Safety Science, 2010, 48(10):1248-1262.

Xin S, Zhang L, Jin X, et al. Reconstruction of the fault tree based on accident evolution - ScienceDirect[J]. Process Safety and Environmental Protection, 2019, 121:307-311.
